# Microbial contamination of spittoons and germicidal effect of irradiation with krypton chloride excimer lamps (Far UV-C 222 nm)

Hiroaki Tanimoto[1]*, Yoshimasa Ogawa[2], Takayuki Nambu[3], Toru Koi[2], Hiroyuki Ohashi[2], Toshinori Okinaga[3], Kazuyo Yamamoto[1]

1 Department of Operative Dentistry, Osaka Dental University, Osaka, Hirakata, Japan, 2 Ushio Inc., Chiyoda-ku, Tokyo, Japan, 3 Department of Microbiology, Osaka Dental University, Osaka, Hirakata, Japan

* tanimoto@cc.osaka-dent.ac.jp

**Data Availability Statement:** within the manuscript: All raw sequence data generated during this research are accessible from the DNA

## Abstract

### Background

In dentistry, instruments, appliances, and body fluids such as saliva or blood are possible sources of infection. Although conventional antiseptic procedures effectively prevent infection, spittoons cannot be sanitized between each treated patient and are usually washed only with running water. However, there is currently no fast and efficient disinfection method that can be implemented between treatments. An optically filtered krypton chloride excimer lamp using ultraviolet light (Far UV-C) in the 200–230 nm wavelength range (innocuous to humans) has been recently used as a virus- and bacteria-inactivating technology. This study aimed to identify the bioburden of a dental spittoon and examine the susceptibility of two oral Streptococcus and two Enterococci to 222-nm Far UV-C by irradiating the spittoon with 222 nm Far UV-C for 5 min before evaluating the disinfection effect.

### Methods

Bacterial analysis and real-time polymerase-chain reaction testing was used to confirm the spittoon's biological contamination. Bacterial susceptibility to a 222-nm Far UV-C was determined with a graded dose irradiation test. After each treatment, the spittoon was irradiated with 222-nm Far UV-C for 5 min, and the disinfecting effect was evaluated. Microbial analysis of the spittoon's surface was performed using the Silva database.

### Results

We found that > 97% of the microbes consisted of six bacterial phyla, whereas no viruses were found. *Pseudomonas aeruginosa* was frequently detected. The 1-log reduction value of two oral-derived Streptococci and two Enterococci species at 222-nm Far UV-C was 4.5–7.3 mJ/cm². Exposure of the spittoon to 222-nm Far UV-C at 3.6–13.5 mJ/cm² significantly decreased bacterial counts (p < 0.001).

Data Bank of Japan under the accession number DRR545394-DRR545396. The related data in this research "Disinfection efficacy of Far UV-C irradiation on spittoon" and "Bacterial susceptibility to Far UV-C in vitro" are available on Open Science Framework at the following DOI: 10.17605/OSF.IO/BCG9Z.

**Funding:** The author(s) received no specific funding for this work.

**Competing interests:** The authors have declared that no competing interests exist.

## Conclusions

Irradiation with 222-nm Far UV-C at 3.6–13.5 mJ/cm$^2$ significantly eliminates bacteria in spittoons, even when they are only rinsed with water. Hence, 222-nm Far UV-C irradiation may inhibit the risk of bacterial transmission from droplets in sink surfaces.

## Introduction

The most common routes of pathogen transmission in healthcare settings are through the hands, either through a patient–health worker or patient–patient pathway [1]. Infectious microorganisms can be found in contaminated bodily fluids (saliva, blood), improperly sterilized equipment/instruments, and in the air [2, 3]. However, owing to a lack of time and personnel costs, there are instances where adequate cleaning and disinfection cannot be performed at each patient changeover, such as with the spittoon. In general, spittoons are rinsed with running water; however, this is considered an insufficient hygiene measure for areas in contact with oral rinses with high biohazard material (Fig 1Ai). Ultraviolet (UV) germicidal irradiation is considered as a viable countermeasure [4]. However, no safe and effective UV disinfection technology has been developed yet, making it difficult to use in an inhabited environment.

A virus- and bacteria-inactivating technology that uses UV light (with no wavelengths harmful to humans) in the 200–230 nm wavelength range (Far UV-C) and with an optical filtered krypton chloride excimer lamp has been recently developed [5]. In Far UV-C devices, a wavelength of 222 nm is the most researched for human safety and germicidal efficacy. In safety evaluation, irradiation with 222-nm Far UV-C of a Xeroderma pigmentosum model mouse (more susceptible to UV damage than the wildtype) exerted no carcinogenic or cataract effects on the skin and eyes of the mouse [6]. Further, direct irradiation of normal human skin inactivates indigenous bacteria without causing erythema [7]. Long-term safety for human eyes has also been reported [8]. In addition, the germicidal effects against pathogenic viruses and bacteria, including the new coronavirus, have been confirmed [9, 10]. These characteristics are advantageous in an inhabited environment; thus, the disinfection effect of 222-nm Far UV-C was evaluated in outpatient toilets and medical mobile phones contaminated with methicillin-resistant *Staphylococcus aureus* [11, 12]. Additionally, there are studies about its safety in surgical site infection control and clinical trials on pressure ulcer disinfection with 222 nm Far UV-C [13, 14].

This study aimed to identify the bioburden of a dental spittoon and examine the susceptibility of two oral Streptococcus and two Enterococci to 222-nm Far UV-C. Toward this goal, a spittoon was irradiated with 222-nm Far UV-C for 5 min, and then the disinfecting effect was evaluated.

## Materials and methods

This study was approved by the Ethical Review Board of Osaka Dental University (Approval No.111224-0).

### Biological contamination of spittoon

**Swab sampling.** A microbial analysis of the biological contamination status of a spittoon used during a day of treatment, rinsed only with water from the dental unit, was conducted.

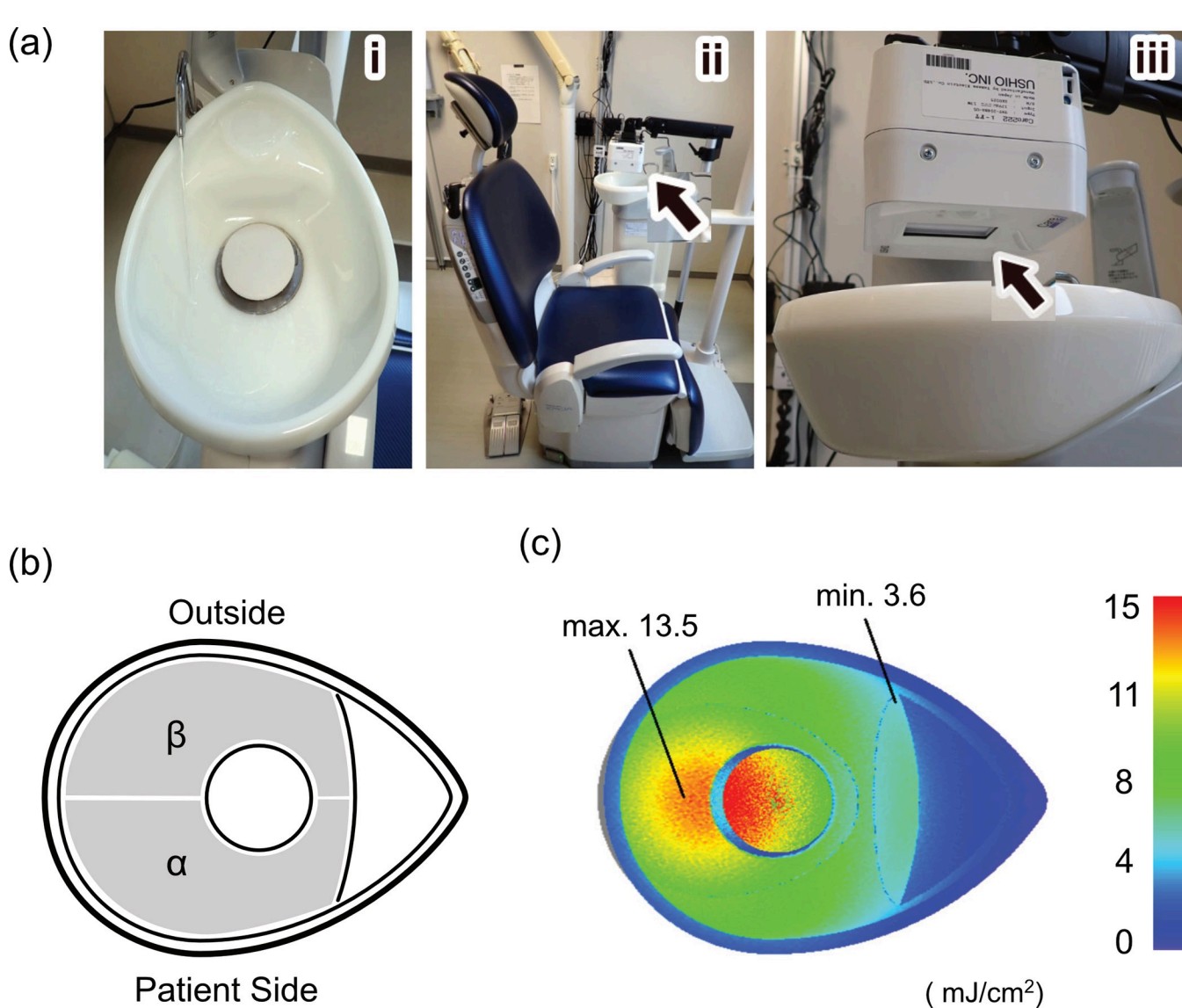

**Fig 1. Protocol of 222-nm Far UV-C irradiation to the spittoon bowl and swab sampling area.** (a) 222-nm Far UV-C irradiation to the spittoon bowl. i: Cleaning of the spittoon bowl with running water; ii: the dental unit and the irradiation lamp unit (arrow); and iii: position of the irradiation window at the irradiation lamp unit (arrow). (b) Swab sampling area. $\alpha$: patient side of 230 cm$^2$, $\beta$: outside area of 230 cm$^2$. (c) Surface irradiation dose of the spittoon bowl. The maximum and minimum irradiation doses in the swab sampling area are 13.5 mJ/cm$^2$ and 3.6 mJ/cm$^2$, respectively.

Severe acute respiratory syndrome coronavirus (SARS-CoV)-2, hepatitis B virus, hepatitis C virus, and human immunodeficiency virus were concurrently detected using real-time polymerase-chain reaction testing. The bacterial and viral samples from the spittoon were collected by wiping the entire surface with a sterile swab (dilution attached swab test phosphate-buffered saline [PBS]; Kanto Chemical Co., Tokyo, Japan) for 20 s after being used by three to five patients in three dental units. Swab samples were immediately frozen at -20°C and transferred to the laboratory.

**DNA extraction, library construction, and high-throughput sequencing.** In a compositional analysis of bacterial communities, the spittoon wipe samples were subjected to DNA extraction and amplicon sequencing as described previously [15, 16]. DNA extraction was performed via chemical and mechanical lysis using Pathogen Lysis Tube S and QIAamp UCP

Pathogen Mini Kit (Qiagen, Hilden, Germany). DNA amplification targeting the V3–V4 region of the 16S ribosomal RNA (rRNA) gene was performed via PCR testing using the primers 341F (5′–TCGTCGGCAGCGTCAGATGTGTATAAGAGACAGCCTACGGGNGGCWGCAG–3′) and 806R (5′–GTCTCGTGGGCTCGGAGATGTGTATAAGAGACAGGGACTACHVGGGTWTCTAAT–3′). PCR amplification was performed using Premix Ex Taq polymerase from Takara Bio (Otsu, Japan), with an initial denaturation at 98°C for 10 s, followed by 26 cycles of denaturation at 98°C for 10 s, annealing at 55°C for 30 s, and extension at 72°C for 1 min. The integrity of the DNA amplicons was confirmed via electrophoresis on 1% agarose gels to detect 460–480 bp fragments. Replicate amplicons were purified using AMPure XP beads from Beckman Coulter (Miami, FL, USA). An additional eight cycles of PCR amplification were performed under the abovementioned conditions to align the purified DNA amplicons at the 3'- and 5'-ends with the adapted primers. The resulting PCR products were purified using AMPure XP beads and quantified using a Quantus fluorometer and a Qubit dsDNA HS Assay Kit (Thermo Fisher, Waltham, MA). The final library was created by pooling the amplicons in equimolar amounts and mixing them with 5% equimolar amounts of PhiX DNA (Illumina). After library construction, 16S rRNA gene sequencing was performed using the Illumina MiSeq platform (Illumina, USA) with $2 \times 250$ bp paired end reads.

**Sequence data processing.** Demultiplexed paired end reads obtained in the sequencing step above were processed using the QIIME 2 (the Quantitative Insights Into Microbial Ecology 2) version 2021.2 pipeline [17]. Using DADA2, the raw reads were filtered for quality and trimmed along with the removal of chimeras based on quality and consensus (via q2-dada2). The following filtering parameters were considered: trim-left-f = 17, trim-left-r = 21, trunc-len-f = 240, and trunc-len-r = 240. The resulting amplicon sequence variants (ASVs) were merged into a single feature table using the q2-feature-table plugin. A naive Bayes classifier (based on q2-feature-classifier) [18] was trained based on the V3-V4 regions of the 16S rRNA sequences from SILVA [19] or the expanded human oral microbiome database (eHOMD; v.15.2) [20] and applied to each amplicon sequence variant for classification. We performed multiple sequence alignment of all ASVs using MAFFT (via q2-alignment). Following paired-end assembly, mass filtering, and removal of chimeras, a total of 71,094 sequences were clustered into 536 identified exact ASVs, both when using SILVA and eHOMD.

**Virus detection.** Viral RNA samples were extracted from a swab sample with QIAamp Viral RNA Mini Kit (Qiagen, Hilden, Germany); then, SARS-CoV-2 was detected with SARS-CoV-2 Detection RT-qPCR Kit for Wastewater (Takara Bio Inc., Shiga, Japan). Hepatitis B virus, hepatitis C virus, and human immunodeficiency virus were detected with a Virus Test Kit Ver.2 (Takara Bio Inc., Shiga, Japan).

## Bacterial susceptibility to Far UV-C *in vitro*

The sensitivity of Streptococci (*Streptococcus salivarius* (NBRC13956), *S. mutans* (NBRC13955)) and Enterococci (*Enterococcus faecalis* (NBRC100480), *E. faecium* (NBRC100485)) to 222-nm Far UV-C were determined as follows. First, using a NanoDrop One (Microvolume UV-Vis Spectrophotometers) and four types of bacteria cultured in modified GAM broth (NISSUI PHARMACEUTICAL CO., LTD., Tokyo, Japan), turbidity (O.D. value) was measured every 2 h of shaking to investigate the logarithmic phase. The measurement wavelength was 600 nm. Colonies of four types of bacteria cultured in modified GAM plate medium were collected from petri dishes, placed in 15-mm centrifuge tubes containing 4 mL modified GAM liquid medium, cultured with shaking until the respective logarithmic growth phases were reached, and diluted with PBS to a countable concentration. Subsequently, the diluted solution was placed in a 35-mm Petri dish and irradiated with a 222-nm far UV-C

light source (Care222® U3 type: TRT-104C11-UI-U3-Z3, Ushio Inc., Tokyo, Japan) with an irradiance of 50 µW/cm². Each strain (n = 3) was irradiated with 5, 10, 20, and 30 mJ/cm². Thereafter, Mitis Salivarius (MS) agar (Sigma-Aldrich Japan G.K.) was inoculated then cultured at 37˚C for 48 h, and the number of colonies was counted in colony forming unit (CFU).

### Disinfection efficacy of Far UV-C irradiation on spittoon

An optical filtered krypton chloride excimer lamp (Care222® i-FT type: TRT-204BS-US; Ushio Inc., Tokyo, Japan) was placed directly above the spittoon, and irradiation was performed for 5 min after patient treatment (Fig 1Aii, 1Aiii). The illuminance of the 222-nm Far UV-C was measured with a Light Meter Spittoon, and irradiation with 222-nm Far UV-C at 3.6–13.5 mJ/cm² significantly eliminated bacteria even when it was washed only with water. These results suggest that 222-nm Far UV-C irradiation may inhibit the pathway of bacterial transmission by droplets from sink surfaces. From the measured irradiance, the irradiance of the entire inner surface of the sink (µW/s) was determined with the non-sequential ray tracing method using illumination optical analysis software (ASAP Next Gen V1 SP3, Global DB Version:6.0; Breault Research Organization, Inc). The irradiance (µW/s) of the entire inner surface of the sink was determined using the non-sequential ray tracing method, and the irradiance of the entire inner surface of the spittoon per irradiation (mJ/cm²) was calculated.

The reflectance of the sink for 222-nm Far UV-C was set to 0.062. Sampling in the 222-nm Far UV-C irradiation test of the spittoon was performed by wiping the bowl's surface with a sponge cloth (Dilution Attached Swab Test (PBS); Kanto Chemical Company, Tokyo, Japan). The patient side [α] (230 cm²) was swabbed before irradiation, whereas the outer side [β] (230 cm²) was swabbed after irradiation. In addition, a test was conducted in which the swabbing surfaces were exchanged before and after irradiation (Fig 1B). The sponge cloth was thoroughly rubbed and washed in 10 mL PBS (pH 7). Subsequently, 100 µL PBS was inoculated into standard agar medium (Nissui Pharmaceutical Co., Ltd., Tokyo, Japan) and into MS agar plate (n = 2 each). They were then aerobically cultured at 37˚C for 48 h, and the CFU was measured. When more than 300 colonies were detected in the petri dish, the CFU was counted as 300.

**Statistical analysis.** We tested the significant difference in the number of CFUs in each medium when the patient side [α] was wiped before 222-nm Far UV-C irradiation and the outside [β] was wiped after irradiation. We also performed the same test when the wiped sides were switched before and after irradiation. Finally, we tested the significant difference in the number of CFUs before and after irradiation without distinguishing between the wiped sides. Significant differences due to 222-nm Far UV-C irradiation were determined using the Wilcoxon signed-rank test of R version 4.2.2 [21].

## Results

### Biological contamination of spittoon

The analysis performed using the Silva database showed that the composition of the microbiota varied slightly among the bowls; however, in each of the three samples, more than 97% were composed of six phyla (Firmicutes, Proteobacteria, Bacteroidota, Actinobacteriota, Fusobacteriota, and Patescibacteria) (Fig 2A). At the genus level, the top 12 genera accounted for more than 70% of the microbiota in each sample (Fig 2B), and 11 genera, except *Pseudomonas*, were endemic to the oral cavity. The species-level analysis conducted using the HOMD database revealed that the top 15 species constituted approximately half of the microbiota composition in each sample (Fig 2C). Among these, 10 species were consistently identified across all three samples, each representing more than 0.5% of the total microbiota composition.

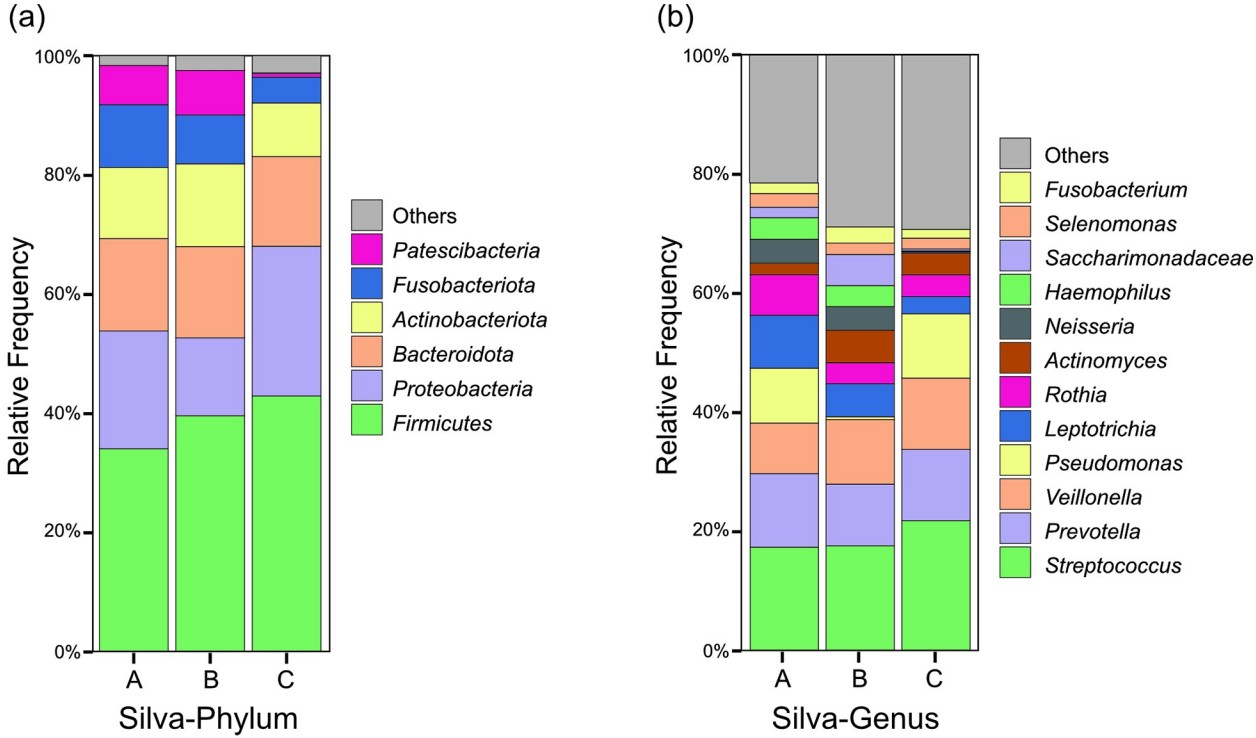

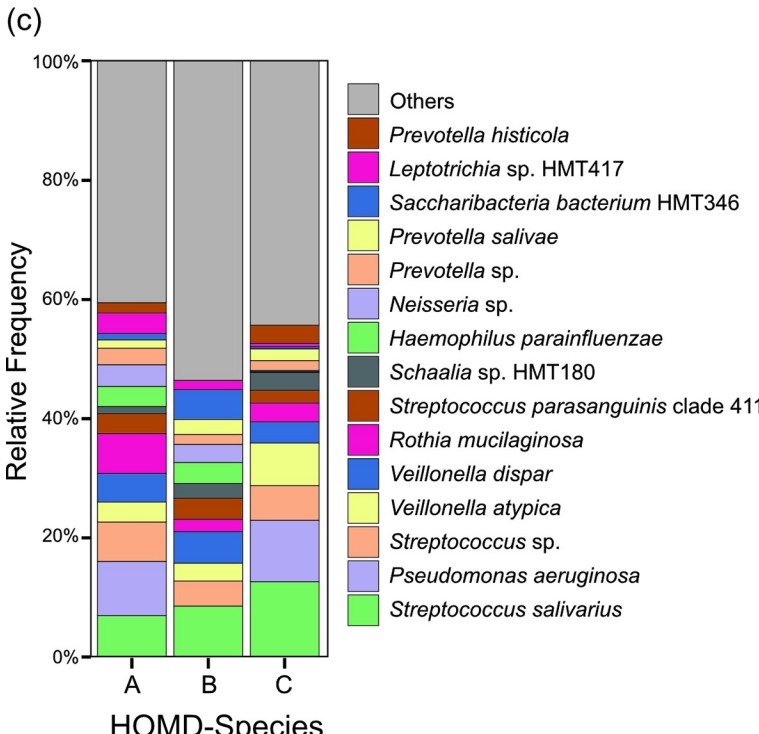

**Fig 2. Compositional analysis of the bacterial communities of the spittoon bawls.** The relative abundance of bacterial communities in each spittoon bawl (A, B, C) is shown at the phylum (a), genus (b), and species (c) levels. The microbiota of each sample is analyzed in the Silva (a, b) or HOMD (c) database.

Meanwhile, various bacterial species from the oral cavity and environment (especially the water system) were detected, and these did not show similar microbiota composition among the three spittoons. The most frequent species were *Streptococcus salivarius* in two samples and *Pseudomonas aeruginosa* in the remaining sample. Meanwhile, none of the four viruses tested were detected.

## Bacterial susceptibility to Far UV-C *in vitro*

According to the approximate formulas obtained from the inactivation rates (n = 3) measured at each 222-nm Far UV-C irradiation dose, the 1-log reduction doses of *S. salivarius* and *S. mutans* were 5.2 mJ/cm$^2$ and 4.8 mJ/cm$^2$, respectively (Fig 3A). Similarly, the 1-log reduction doses of *E. faecium* and *E. faecalis* were 4.5 mJ/cm$^2$ and 7.3 mJ/cm$^2$, respectively (Fig 3B).

## Disinfection efficacy of Far UV-C irradiation on spittoon

Irradiation by setting the light source unit at a distance of 5 cm directly above the spittoon was performed, and the minimum and maximum irradiation doses on the spittoon surface used for wiping were 3.6 and 13.5 mJ/cm$^2$, respectively (Fig 1C). The CFU of the spittoon surface on the patient side before irradiation [α] and on the outside after irradiation [β] (Fig 1B) were significantly reduced after irradiation under the culture conditions of both standard medium (n = 33) and MS medium (n = 30) (both p < 0.001). Similarly, when comparing CFUs on the outside before irradiation [β] and on the patient side after irradiation [α] (Fig 1B), the CFUs were significantly reduced after irradiation under the culture conditions of both standard medium (n = 32) and MS medium (n = 30) (both p < 0.001) (Fig 4A). Next, all CFUs before and after irradiation under standard medium (n = 65) and MS medium (n = 60) were compared. The mean CFU before irradiation was 53.5 in the standard medium, and this significantly decreased to 15 after irradiation (p < 0.001). Similarly, the mean CFU before irradiation was 13 in the MS medium, and this significantly decreased to 1.2 after irradiation (p < 0.001) (Fig 4B, Table 1).

## Discussion

The sink, as a part of the water system in hospitals, is an established reservoir for nosocomial infection of *P. aeruginosa* [22, 23]. There have also been reports of *P. aeruginosa* detection from dental unit water systems [24]. In this study, *P. aeruginosa* was detected at a high frequency from all the analyzed spitting bowls using microbial community analysis, suggesting that spitting bowls could also serve as reservoirs for infection. In vitro studies have shown that exposure to 6 mJ/cm$^2$ of 222 nm far UV-C reduced the bacterial population of *P. aeruginosa* by 100- to 1000-fold [10], indicating that irradiation with 222 nm Far UV-C can be used to disinfect *P. aeruginosa* on the surface of spitting bowls (Figs 1 and 4). The decrease in the bacterial count on standard medium indicated the disinfection effect against mesophilic aerobic bacteria. Meanwhile, the evaluation in MS medium mainly shows a decrease in oral streptococci, which are of oral origin (Table 1, Fig 4). Further, 222-nm Far-UVC irradiation of 3.6–13.5 mJ/cm$^2$ reduced mesophilic aerobic bacteria to less than one-third and oral streptococci to less than one-tenth in the spittoons (Table 1). Bacteria on sink surfaces spread around with water droplets splashing from the sink [25]. The bacterial disinfection of this study suggests that disinfection with 222-nm Far-UVC can be a method to suppress the risk of bacterial transmission from the sink (Fig 4). Furthermore, because 222-nm Far UV-C irradiation can be applied in a human environment, sinks (and their surroundings) can be irradiated even during treatment, and increasing the irradiation dose is expected to improve the disinfection effect.

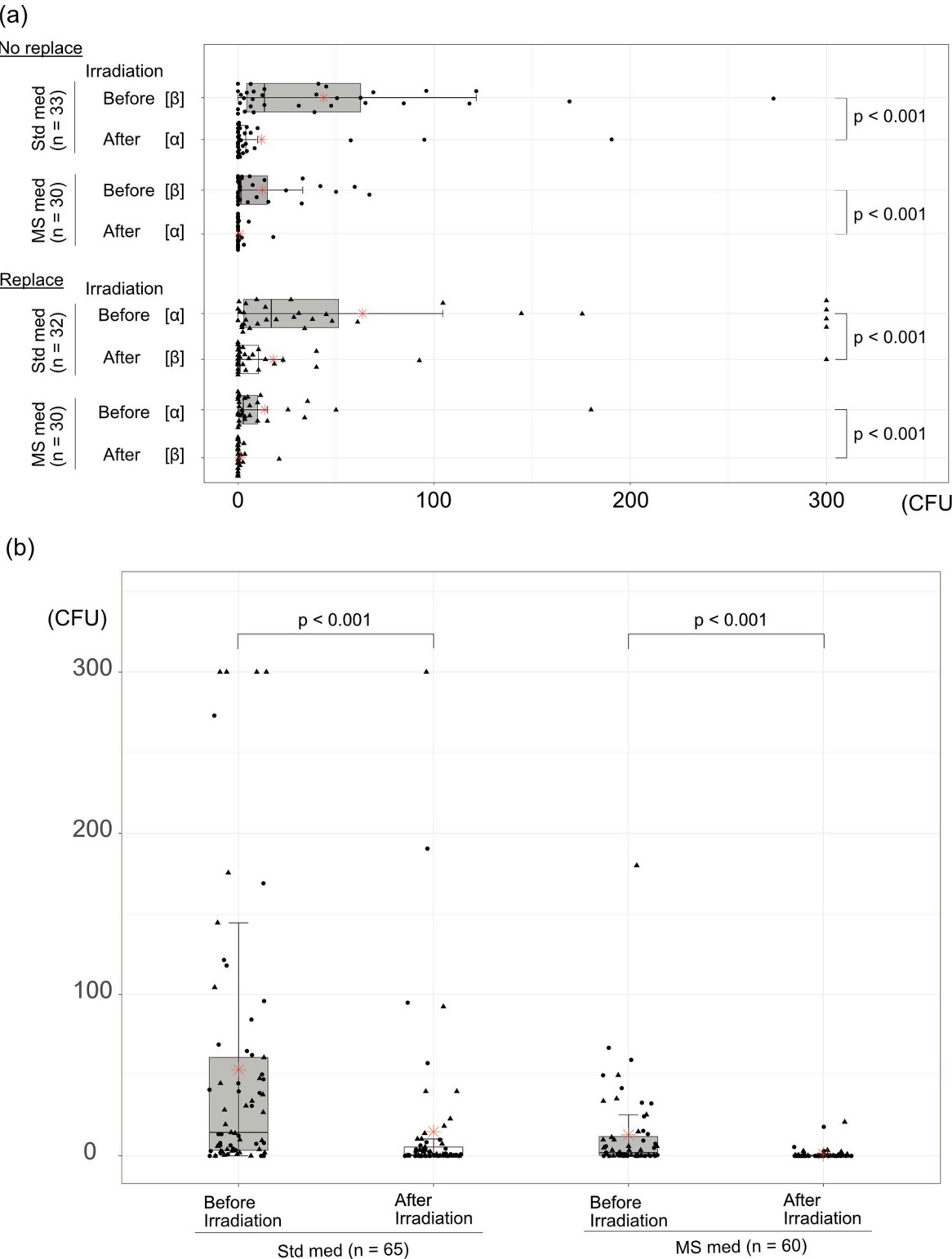

**Fig 3. 222 nm Far UV-C susceptibility for Streptococcus and Enterococci.** (a) Streptococcus, (b) Enterococci Reduction = $N/N_0$, where N is the number after irradiation with 222-nm Far-UVC, and $N_0$ is the number before irradiation.

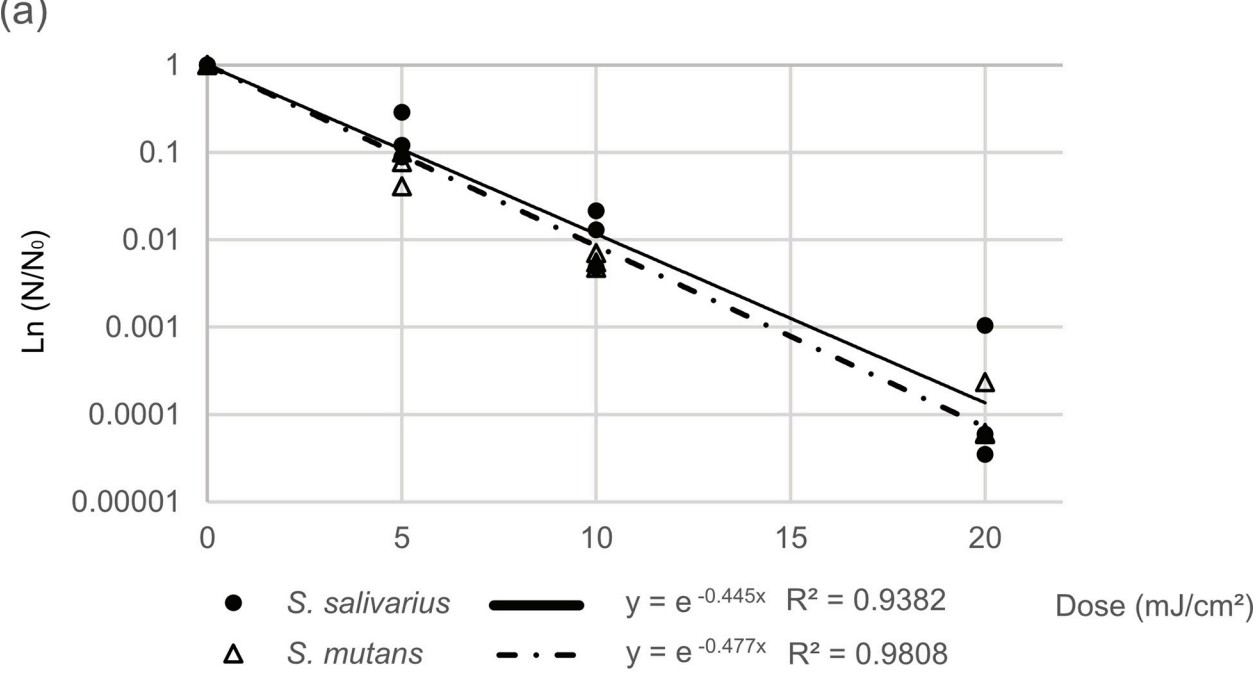

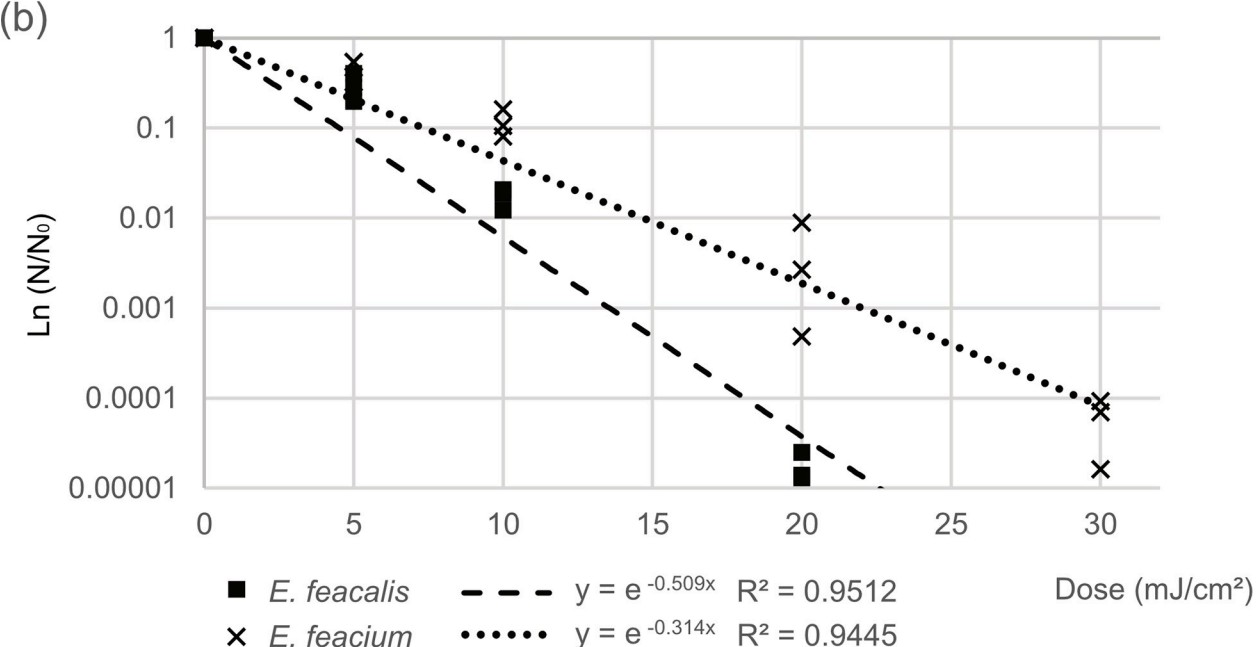

**Fig 4. Reduction of bacterial colony forming unit (CFU) on the spittoon bowl surface by 222-nm Far UV-C irradiation.** (a) Replacement of the wiping surface before and after irradiation. α is the patient side, and β is outside of the spittoon bowl. (b) All CFUs before and after irradiation. The red asterisk indicates the average number of CFU. ●: CFU number before irradiation at β and after irradiation at α as the "No replacement" of the swab sampling area. ▲: CFU number before irradiation at α and after irradiation at β as the "Replace" of the swab sampling area. Std med: standard medium. MS med: Mitis Salivarius medium.

**Table 1. Reduction of the bacterial colony forming unit on the dental unit spittoon surface by 222-nm Far UV-C irradiation.**

| | | Mean | Median | SD | SE | 95% IC |
|---|---|---|---|---|---|---|
| **Standard medium (n = 65)** | | | | | | |
| | Before irradiation | 53.5 | 14.5 | 81.5 | 10.1 | 19.8 |
| | After irradiation | 15.0 | 0.5 | 46.4 | 5.7 | 11.3 |
| **Mitis Salivarius medium (n = 60)** | | | | | | |
| | Before irradiation | 13.0 | 2 | 27.4 | 3.5 | 6.9 |
| | After irradiation | 1.2 | 0 | 3.6 | 0.5 | 0.9 |

Unit: CFU

SD: standard deviation, SE: standard error, 95% CI: 95% confidence interval

## Conclusions

Irradiation with 222-nm Far UV-C at 3.6–13.5 mJ/cm$^2$ significantly eliminates bacteria in spittoons, even when they are only washed with water. These results suggest that 222-nm Far UV-C irradiation may inhibit the droplet pathway of bacterial transmission risk from sink surfaces.

## Acknowledgments

The authors would like to express their gratitude to Mr. Atsushi Honda of Ushio Inc. for calculating the irradiation dose on the surface of the spittoon bowl based on the measured illuminance. We would also like to thank Editage (www.editage.com) for English language editing.

## Author Contributions

**Conceptualization:** Yoshimasa Ogawa, Takayuki Nambu, Toru Koi, Hiroyuki Ohashi, Toshinori Okinaga, Kazuyo Yamamoto.

**Data curation:** Yoshimasa Ogawa, Takayuki Nambu.

**Formal analysis:** Yoshimasa Ogawa, Takayuki Nambu, Toru Koi, Hiroyuki Ohashi, Toshinori Okinaga, Kazuyo Yamamoto.

**Investigation:** Yoshimasa Ogawa, Takayuki Nambu, Toru Koi, Hiroyuki Ohashi, Toshinori Okinaga, Kazuyo Yamamoto.

**Methodology:** Yoshimasa Ogawa, Takayuki Nambu, Toru Koi, Hiroyuki Ohashi, Toshinori Okinaga, Kazuyo Yamamoto.

**Supervision:** Toshinori Okinaga, Kazuyo Yamamoto.

**Writing – original draft:** Hiroaki Tanimoto, Yoshimasa Ogawa, Takayuki Nambu, Toru Koi.

**Writing – review & editing:** Hiroaki Tanimoto, Yoshimasa Ogawa, Takayuki Nambu, Toru Koi, Hiroyuki Ohashi, Toshinori Okinaga, Kazuyo Yamamoto.

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
