## [Decision Letter · Decision Letter 0]

4 Feb 2024

PONE-D-23-38195Microbial contamination of spittoons and germicidal effect of irradiation with Krypton Chloride excimer lamps (Far UV-C 222 nm)PLOS ONE

Dear Dr. TANIMOTO,

Thank you for submitting your manuscript to PLOS ONE. After careful consideration, we feel that it has merit but does not fully meet PLOS ONE’s publication criteria as it currently stands. Therefore, we invite you to submit a revised version of the manuscript that addresses the points raised during the review process.

**ACADEMIC EDITOR: **Please kindly attend to comments below. Thank you

We look forward to receiving your revised manuscript.

Kind regards,

Charles Odilichukwu R. Okpala

Academic Editor

PLOS ONE

Journal Requirements:

3. We note that your Data Availability Statement is currently as follows: within the manuscript and/or Supporting Information files

5. Please amend the manuscript submission data (via Edit Submission) to include author Yoshimasa Ogawab, Takayuki Nambuc, Toru Koib, Toshinori Okinagac and Kazuyo Yamamotoa.

**Additional Editor Comments:**

Please, authors kindly attend to the reviewers' comments

Reviewers' comments:

Reviewer's Responses to Questions

**Comments to the Author**

1. Is the manuscript technically sound, and do the data support the conclusions?

Reviewer #1: Yes

Reviewer #2: Partly

2. Has the statistical analysis been performed appropriately and rigorously? 

Reviewer #1: Yes

Reviewer #2: I Don't Know

3. Have the authors made all data underlying the findings in their manuscript fully available?

Reviewer #1: Yes

Reviewer #2: No

4. Is the manuscript presented in an intelligible fashion and written in standard English?

Reviewer #1: No

Reviewer #2: No

5. Review Comments to the Author

Reviewer #1: a. The study and findings are quite interesting. However, the paper would be easier to read if both the Methodology and Results are arranged in subsections; for example, the methodology could have subsections such as Collection of samples, determination of microbial load, identification of contaminants etc.

b. There is a question in the methodology section that needs to be answered.

Reviewer #2: Title: Microbial contamination of spittoons and germicidal effect of irradiation with Krypton Chloride excimer lamps (Far UV-C 222 nm)

Comments to the authors

I appreciate the author's hard effort.

The authors measure the disinfectant effect of 222 nm Far UV-C radiation on dental spittoon, one may determine the bioburden of the spittoon and assess the sensitivity of two oral Streptococcus and two Enterococci to the radiation.

Although the study presented here is good, several crucial data points are missing.

Major concern

1. The main issue is that each method is included in a single paragraph. It ought to be divided into multiple sections. Each technique must to have a unique the title. Every protocol should be written in a single, concise sentence. Every protocol should be thoroughly written

2. Did you create a growth curve for line 105, "First, each strain was grown to the mid-logarithmic growth phase"? How did you find out each strain's mid-logarithmic development phase?

3. The 16s rRNA sequencing technique is absent. This is a significant experiment; could you further elaborate on the 16s rRNA sequencing methodology?

4. The manuscript needs to include the steps for phylogenetic diversity, OTU grouping and taxonomical assignment, trimmed sequence, and data sequence filtering in QIIME 2.

5. How did you determine that the change is significant? The paper has to include alpha and beta diversity analysis along with figures. How many germs are extracted from the sample and how many are common? There should be a Venn diagram.

Minor Comments

- Please add reference to line 43-44, 50- 51

- Italicize the bacterial name as in line 139 “Pseudomonas,” Pls check the whole manuscript.

- Line 84, change “was attempted” to “ was conducted”

- Line 93, what is the size of amplified fragment? Add primers sequences used?

- Line 106, “and suspended in PBS? Is not clear, GAM broth or PBS?. Write the whole protocol in details.

- - Lines 107–109 This procedure is quite brief. "How did you perform the irradiation? Could you perhaps elaborate on the protocol in a separate paragraph?

- - Lines 109–110 The CFU method should be described in detail as well.

- Line 115 “Our results suggest that 222 nm Far UV-C irradiation may inhibit the pathway of bacterial transmission by droplets from sink surfaces.” Should be moved to result.

- Line 130, “number of bacteria” Do you mean CFU? Why did you count 300 colonies when there were actually more?

- Line 141-142 “Ten of these species were present in three samples with more than 0.5% in common” Is it among the 15 sample or among the whole species numbers? You need to explain or justify why it precent in very low percentage.

- Enter the complete names of the SD, SE, and IC below the Table 1 as short notes. Include the "CFU" in the table itself as well.

-A native English speaker should review the paper. Please review the entire document.

6. PLOS authors have the option to publish the peer review history of their article (what does this mean?). If published, this will include your full peer review and any attached files.

Reviewer #1: **Yes: **Ifeoma Maureen Ezeonu, Ph.D.

Reviewer #2: **Yes: **Anis Rageh Al-Maleki

---

## [Author Response · Author response to Decision Letter 0]

16 May 2024

Dr. Charles Odilichukwu R. Okpala

PLOS ONE

Dear Dr. Okpala,

We wish to re-submit the manuscript titled “Microbial contamination of spittoons and germicidal effect of irradiation with Krypton Chloride excimer lamps (Far UV-C 222 nm).” The manuscript ID is PONE-D-23-38195.

We thank you and the reviewers for your thoughtful suggestions and insights. The manuscript has benefited from these insightful suggestions. I look forward to working with you and the reviewers to move this manuscript closer to publication in PLOS ONE.

The manuscript has been rechecked and the necessary changes have been made in accordance with the reviewers’ suggestions. The responses to all comments have been prepared and provided below. The changes in the manuscript are in red text.

Thank you for your consideration. I look forward to hearing from you.

Sincerely,

Hiroaki Tanimoto

Address: 1-5-17 Otenae, Chuo-ku, Osaka 540-0008, Japan

Email: tanimoto@cc.osaka-dent.ac.jp

---

## [Decision Letter · Decision Letter 1]

10 Jun 2024

PONE-D-23-38195R1Microbial contamination of spittoons and germicidal effect of irradiation with Krypton Chloride excimer lamps (Far UV-C 222 nm)PLOS ONE

Dear Dr. TANIMOTO,

Thank you for submitting your manuscript to PLOS ONE. After careful consideration, we feel that it has merit but does not fully meet PLOS ONE’s publication criteria as it currently stands. Therefore, we invite you to submit a revised version of the manuscript that addresses the points raised during the review process.

**ACADEMIC EDITOR: **Please kindly see comments below.==============================

We look forward to receiving your revised manuscript.

Kind regards,

Charles Odilichukwu R. Okpala

Academic Editor

PLOS ONE

Journal Requirements:

Additional Editor Comments:

I have critically examined the revised manuscript, and authors have addressed critical comments of the work.

Authors, kindly attend to the following and make sure to only upload the final draft. Use red font to indicate the revised areas.

a) Kindly include in your introduction about the global ramifications of microbial contamination in clinical settings - mortality, etc, bring in published data showing different continents, narrow down to Asia, then one in Japan. (this should be your paragraph 1). Make sure to contextualize the paragraph 1 to the subject of this work.

b) In materials and methods, please separate out the data analysis aspects and create the last subsection "statistical analysis", and have all of them there.

c) Your discussion needs additional depth. Please, provide deeper explanation of the results, not just about 'what', but more about 'how' and 'why?' . Please also, kindly make sure you include (Refer to Table 1) (Refer to Figure 2) (Refer to Figure 3) (Refer to Figure 4) in all the places where data of each has been discussed. So, all the Tables/Figures in the results, must be captured in the discussion

The editor will thoroughly examine these in your revised manuscript. Thank you very much and look forward to your revised manuscript.

Reviewers' comments:

Reviewer's Responses to Questions

**Comments to the Author**

1. If the authors have adequately addressed your comments raised in a previous round of review and you feel that this manuscript is now acceptable for publication, you may indicate that here to bypass the “Comments to the Author” section, enter your conflict of interest statement in the “Confidential to Editor” section, and submit your "Accept" recommendation.

Reviewer #1: All comments have been addressed

2. Is the manuscript technically sound, and do the data support the conclusions?

Reviewer #1: Yes

3. Has the statistical analysis been performed appropriately and rigorously? 

Reviewer #1: Yes

4. Have the authors made all data underlying the findings in their manuscript fully available?

Reviewer #1: Yes

5. Is the manuscript presented in an intelligible fashion and written in standard English?

Reviewer #1: Yes

6. Review Comments to the Author

Reviewer #1: Although, one would have wished for the use of a larger sample size in this study, the paper can be accepted because of the detailed analysis of the few samples employed. Moreover, the paper introduces a novel method of reducing contamination and transmission of infections via reusable dental instruments.

7. PLOS authors have the option to publish the peer review history of their article (what does this mean?). If published, this will include your full peer review and any attached files.

Reviewer #1: **Yes: **Ifeoma Maureen Ezeonu

---

## [Author Response · Author response to Decision Letter 1]

20 Jul 2024

Dear Dr. Charles Odilichukwu R. Okpala

Academic Editor

PLOS ONE

Thank you for giving me the opportunity to submit a revised draft of my manuscript titled “Microbial contamination of spittoons and germicidal effect of irradiation with Krypton Chloride excimer lamps (Far UV-C 222 nm)” to PLOS ONE. We appreciate the time and effort that you have dedicated to providing your valuable feedback on my manuscript. 

I am/We are grateful to your insightful comments on my paper. We have been able to incorporate changes to reflect some of the suggestions provided. We have highlighted the changes within the manuscript.

Editor Comments

Comment a) Kindly include in your introduction about the global ramifications of microbial contamination in clinical settings - mortality, etc, bring in published data showing different continents, narrow down to Asia, then one in Japan. (this should be your paragraph 1). Make sure to contextualize the paragraph 1 to the subject of this work.

Response: Thank you for pointing that out. The focus of our study was not on survival rates or mortality rates in hospital-acquired infections, but on reducing biological contamination in dental treatment booths during dental treatment, so we were unable to obtain data on research from this perspective. On the other hand, there were studies on the mortality rates of hospital-acquired infections worldwide by pathogenic bacteria, but these were not directly linked to the target of this study, which was to reduce biological contamination.

Comment b) In materials and methods, please separate out the data analysis aspects and create the last subsection "statistical analysis", and have all of them there.

Response: Thank you for your suggestions to make the clearer. We have created a subsection "Statistical Analysis" in "Materials and Methods" to summarize the data. [Line 170 - 176]

Comment c) Your discussion needs additional depth. Please, provide deeper explanation of the results, not just about 'what', but more about 'how' and 'why?' . Please also, kindly make sure you include (Refer to Table 1) (Refer to Figure 2) (Refer to Figure 3) (Refer to Figure 4) in all the places where data of each has been discussed. So, all the Tables/Figures in the results, must be captured in the discussion

Response: Thank you for your comments that deepened the discussion appropriately. I have added missing figures and tables [Line 245, 247, 252] and incorporated all figures and tables in the results into the discussion. Also, to clarify the relationship between existing reports and our results [why], I have clarified the amount of Far UV-C irradiation and the inactivation amount of Pseudomonas aeruginosa in the existing results. [Line 242-243] On the other hand, I have deleted the phrase "which were only rinsed with flowing water," which is not directly related to the comparison of irradiation and non-irradiation [Line 249].

---

## [Editor Report · Decision Letter 2]

24 Jul 2024

Microbial contamination of spittoons and germicidal effect of irradiation with Krypton Chloride excimer lamps (Far UV-C 222 nm)

PONE-D-23-38195R2

Dear Dr. TANIMOTO,

We’re pleased to inform you that your manuscript has been judged scientifically suitable for publication and will be formally accepted for publication once it meets all outstanding technical requirements.

Kind regards,

Charles Odilichukwu R. Okpala

Academic Editor

PLOS ONE

Additional Editor Comments (optional):

I have critically evaluated the revised manuscript, as well as the responses reviewers provided, and very satisfied to accept it for publication.
---

## [Editor Report · Acceptance letter]

29 Jul 2024

PONE-D-23-38195R2 

PLOS ONE

Dear Dr. TANIMOTO, 

I'm pleased to inform you that your manuscript has been deemed suitable for publication in PLOS ONE. Congratulations! Your manuscript is now being handed over to our production team.

Kind regards, 

on behalf of

Dr. Charles Odilichukwu R. Okpala 

Academic Editor

PLOS ONE